# The Impact of Drug–Drug Interactions on the Toxicity Profile of Combined Treatment with BRAF and MEK Inhibitors in Patients with BRAF-Mutated Metastatic Melanoma

**DOI:** 10.3390/cancers15184587

**Published:** 2023-09-15

**Authors:** Silvia Mezi, Andrea Botticelli, Simone Scagnoli, Giulia Pomati, Giulia Fiscon, Federica De Galitiis, Francesca Romana Di Pietro, Sofia Verkhovskaia, Sasan Amirhassankhani, Simona Pisegna, Giovanna Gentile, Maurizio Simmaco, Bjoern Gohlke, Robert Preissner, Paolo Marchetti

**Affiliations:** 1Department of Radiological, Oncological, and Anatomopathological Sciences, Sapienza University of Rome, 00161 Rome, Italy; silvia.mezi@uniroma1.it (S.M.); andrea.botticelli@uniroma1.it (A.B.); 2Department of Molecular Medicine, Sapienza University of Rome, 00161 Rome, Italy; giulia.pomati@uniroma1.it (G.P.); simona.pisegna@uniroma1.it (S.P.); 3Department of Computer, Control, and Management Engineering “Antonio Ruberti”, Sapienza University of Rome, 00161 Rome, Italy; giulia.fiscon@uniroma1.it; 4Istituto Dermopatico dell’Immacolata, IDI-IRCCS, 00144 Rome, Italy; degalitiis@yahoo.it (F.D.G.); francescadipietro27@gmail.com (F.R.D.P.); sofiglia@gmail.com (S.V.); paolo.marchetti@uniroma1.it (P.M.); 5Department of Urology, S. Orsola-Malpighi Hospital, University of Bologna, Via Palagi, 40126 Bologna, Italy; sasan.am@gmail.com; 6Department of Neuroscience, Mental Health, and Sensory Organs (NESMOS), Faculty of Medicine and Psychology, Sapienza University, 00185 Rome, Italy; giovanna.gentile@uniroma1.it (G.G.); maurizio.simmaco@uniroma1.it (M.S.); 7Unit of Laboratory and Advanced Molecular Diagnostics, ‘Sant’Andrea’ University Hospital, 00189 Rome, Italy; 8Structural Bioinformatics Group, Institute for Physiology, Charité-University Medicine Berlin, 10117 Berlin, Germany; bjoern.gohlke@charite.de (B.G.); robert.preissner@charite.de (R.P.)

**Keywords:** BRAF inhibitors, MEK inhibitors, BRAF mutation, metastatic melanoma, drug–drug interaction, cardiological toxicity

## Abstract

**Simple Summary:**

Drug–drug interactions (DDIs) risk is quite common, potentially significant, and often underestimated. In this context, an advanced DDI detection software (Drug-PIN, V. 2/23) was used to assess the DDIs in a retrospective cohort of 177 patients with metastatic BRAF-mutated cutaneous melanoma treated with BRAF and MEK inhibitors. Furthermore, we evaluated the impact of DDIs on the toxicity profile. Here, we report that the median Drug-PIN score significantly increased when the target combination was added to the patient’s home therapy (*p*-value < 0.0001). Moreover, DDIs emerged as a critical issue for the risk of treatment-related cardiovascular toxicity. Both Drug-PIN score (*p* = 0.0291) and traffic light (*p* = 0.00821) were significant predictors of cardiotoxicity onset. Our results suggest evaluating DDIs in the clinical practice of melanoma patients treated with BRAF/MEK inhibitors to reduce potentially avoidable toxicities and improve treatment tolerability and patients’ quality of life.

**Abstract:**

Background: BRAF and MEK inhibition is a successful strategy in managing BRAF-mutant melanoma, even if the treatment-related toxicity is substantial. We analyzed the role of drug–drug interactions (DDI) on the toxicity profile of anti-BRAF/anti-MEK therapy. Methods: In this multicenter, observational, and retrospective study, DDIs were assessed using Drug-PIN software (V 2/23). The association between the Drug-PIN continuous score or the Drug-PIN traffic light and the occurrence of treatment-related toxicities and oncological outcomes was evaluated. Results: In total, 177 patients with advanced BRAF-mutated melanoma undergoing BRAF/MEK targeted therapy were included. All grade toxicity was registered in 79% of patients. Cardiovascular toxicities occurred in 31 patients (17.5%). Further, 94 (55.9%) patients had comorbidities requiring specific pharmacological treatments. The median Drug-PIN score significantly increased when the target combination was added to the patient’s home therapy (*p*-value < 0.0001). Cardiovascular toxicity was significantly associated with the Drug-PIN score (*p*-value = 0.048). The Drug-PIN traffic light (*p* = 0.00821) and the Drug-PIN score (*p* = 0.0291) were seen to be significant predictors of cardiotoxicity. Patients with low-grade vs. high-grade interactions showed a better prognosis regarding overall survival (OS) (*p* = 0.0045) and progression-free survival (PFS) (*p* = 0.012). The survival analysis of the subgroup of patients with cardiological toxicity demonstrated that patients with low-grade vs. high-grade DDIs had better outcomes in terms of OS (*p* = 0.0012) and a trend toward significance in PFS (*p* = 0.068). Conclusions: DDIs emerged as a critical issue for the risk of treatment-related cardiovascular toxicity. Our findings support the utility of DDI assessment in melanoma patients treated with BRAF/MEK inhibitors.

## 1. Introduction

Cutaneous melanoma (M) represents 1.7% of all global cancer diagnoses [1]. Over the last 10 years, M incidence has consistently risen. However, the M-specific mortality decreased by more than 30% due to the introduction of novel therapeutic agents and their use in combinations [2]. Approximately half of the patients with M harbor an activating mutation in the serine-threonine kinase BRAF, 90% of which occur at codon 600 in exon 15 V600E (substituting valine to glutamine) [3]. Other BRAF-activating mutations are rare [4]. BRAF acts via the RAS/MAPK pathway, which regulates cell proliferation, differentiation, migration, and apoptosis. BRAF mutation (BRAFv600) activates BRAF and upregulates the downstream signal transduction in the MAP kinase pathway involving different mechanisms of M carcinogenesis, growth, progression, and immune escape [5,6,7]. The BRAF mutation is associated with reduced survival compared to wild-type melanoma [8]. Targeting BRAF, with the small selective molecule inhibitors vemurafenib (V), dabrafenib (D), and encorafenib (E), showed clinical efficacy and improved survival outcomes in untreated BRAFV600 metastatic melanomas (MM) compared to standard chemotherapy [9,10,11,12]. Despite encouraging initial response rates, almost 50% of patients treated with BRAF inhibitors (BRAFi) monotherapies relapse within six months, with a median progression-free survival (PFS) from 5 to 10 months in landmark phase III trials (BRIM-3, BREAK-3) [11,12]. Failure of treatment with BRAFi is related to a paradoxical hyperactivation of the MEK-mediated signaling cascade, leading to the rapid development of drug resistance [13,14,15,16,17]. In addition, using BRAFi may result in the development of secondary early squamous cell carcinomas induced by the paradoxical activation of the MAP kinase pathway occurring in non-cancer cells in which the oncogenic BRAFV600 is lacking [18,19]. Inhibitors of MEK (MEKi), a signaling molecule downstream of BRAF, when co-administrated with BRAFi, were demonstrated to dramatically enhance BRAFi activity and delay the development of biological resistance [20,21,22,23]. Considering this synergistic effect, the oral small molecules trametinib (T), cobimetinib (C), and binimetinib (B) have been investigated in association with BRAFi in different first-line phase III trials enrolling untreated MM patients with V600 mutation [20,21,22,23]. The D+T combination showed superior efficacy to D and V monotherapies in COMBI-D and COMBI-V trials, respectively [24,25]. Likewise, in the coBRIM trial, V+C improved PFS, overall survival (OS), and response rate versus V alone [22,26]. Subsequently, a phase III study (COLUMBUS) compared E+B vs. single agents in patients with BRAFV600 MM who had not received prior therapies or had progressed on or after previous first-line immunotherapy [23,27]. BRAFi/MEKi combination therapy has been shown to have superior PFS and OS compared to V alone. At the same time, the improvements in survival outcomes did not reach statistical significance when BRAF/MEK association was compared to E monotherapy [28]. Treatment with BRAFi/MEKi was associated with adverse events (AEs) of any grade in almost all patients (>90%) across phase III trials. Grade 3–4 AEs occurred in 46–56% of patients treated with D-T (COMBI-D, COMBI-V) and 69% of patients undergoing V plus C (coBRIM), respectively. Furthermore, 58% of pts in the E-B arm in the COLUMBUS study (part I) experienced a severe AE (G3–4) [29]. BRAFi/MEKi combinations shared similar “class effects” including gastrointestinal toxicity, impaired liver function, and skin toxicity [29]. On the other hand, in addition to squamous cell carcinoma, arthralgia was a specific BRAFi class reaction. At the same time, ocular edema and cardiovascular toxicity were mainly associated with MEKi therapy as well as with induction in almost all treated patients of a papulopustular exanthema.

Neuro-retinal detachment, muscular problems, hypertension, and ventricular ejection fraction decrease were also related to MEK inhibition. Moreover, specific side effects were related to specific combinations. Skin toxicities, including Stevens–Johnson syndrome, photosensitivity, and acute kidney injury, were higher using the V+C combination. Pyrexia and elevated C-reactive protein were significantly associated with D+T, whereas gastrointestinal AEs, arthralgia, peripheral neuropathies, renal disorder, and an increase in Guillain–Barrè syndrome were significantly associated with E+B [30]. Based on this heavy toxicity profile, proactive management of toxicities is needed to ensure better clinical outcomes avoiding unnecessary treatment discontinuation. Moreover, serious AEs and cumulative toxicities negatively impact the patient’s quality of life [31]. The toxicity spectrum of the combo BRAFi/MEKi treatment can potentially be even worse for all pharmacokinetic interactions related to the intake of foods, dietary supplements, complementary alternative therapies, excipients, and drugs that can interfere with the pharmacodynamics and pharmacokinetics of the BRAFi/MEKi treatment combinations [32]. Environmental factors also affect the absorption, distribution, metabolism, and excretion of the drug, which may also be affected by interpatient variability, given the potential impact of age, gender, genetics, and comorbidity conditions on drug handling [33,34]. In this complex scenario, drug–drug interactions (DDIs), inducing metabolic interference, can result in drug toxicities, reduced pharmacological effects, and adverse drug reactions. DDIs can substantially influence the drug activity of BRAFi/MEKi anticancer therapy in either a beneficial or detrimental manner, reducing treatment efficacy or promoting the development of adverse drug reactions [35]. Drugs undergo multiple metabolic pathways and modifications, which could affect plasma drug concentrations [36]. DDIs usually involve all cytochromes of the P450 (CYPs) enzyme superfamily, P-glycoprotein, ATP-binding cassette transporters, as well as detoxifying and DNA-repair enzymes, fundamental in drug metabolism and central to the occurrence of drugs interaction [37,38,39,40,41,42,43,44].

Although the risk of DDI is potentially significant and quite common, DDIs in cancer treatment were investigated only in a few retrospective studies in which no patients with MM treated with target therapies were included [45,46,47]. This multicentric retrospective study aims to describe the role of DDIs on the toxicity profile and clinical outcomes in a real-world population treated with BRAFi/MEKi therapy using Drug-PIN^®^ (Personalized Interactions Network), a medical software able to detect and improve drug interactions in combination with patient profiles including demographic, clinical and biochemical data [35,48].

The main objective of this study was to retrospectively assess and define the risk of drug interactions in clinical practice and their impact on the toxicity spectrum of patients with BRAFV600 MM treated with BRAFi/MEKi inhibitors, with the intent to reduce potentially avoidable toxicities by improving the tolerability of treatment and the quality of life of patients.

## 2. Materials and Methods

### 2.1. Patients and Treatments

The study is an observational, multicenter, retrospective study, including patients with age 18 years or older and histologically confirmed diagnosis of metastatic/unresectable BRAFV600 CM who received at least 1 month of BRAFi/MEKi combination therapy and had at least one postbaseline safety assessment from January 2018 to October 2021. Clinical data were collected from both Policlinico Umberto I—Sapienza University of Rome and Istituto Dermopatico dell’Immacolata (IDI-IRCCS) of Rome. The study was conducted following the Declaration of Helsinki, and the protocol was approved by the Ethics Committee of the Coordinating Center (Sapienza University of Rome Prot. 0435/2021 Rif. 6332). Advanced disease setting was confirmed with contrast-enhanced computed tomography (CT) scan and, when indicated, magnetic resonance imaging (MRI) and whole body (WB) PET/CT. Patients received a combination of D/T, E/B, or V/C at a standard dose according to local clinical practice until disease progression or unacceptable toxicity occurrence. Dose reduction levels were admitted, reported, and collected.

Full availability of data about patient clinical characteristics, comorbidities, and concomitant medications were needed as additional inclusion criteria. All patients were assessed for safety and treatment outcomes. Toxicities were evaluated according to the parameters provided by the National Cancer Institute Common Toxicity Criteria (CTCAE), version 5.0. PFS was defined as the time from the beginning of target therapy treatment to disease progression or death. OS was defined as the time from the start of treatment to death. Patients without events were considered censored during the last follow-up for PFS and OS. The data cut-off period was June 2022. Written informed consent was obtained from all patients for anonymous clinical data processing for research purposes.

### 2.2. Assessment of Drug Interactions

Drug–drug interactions were assessed using Drug-PIN^®^ software (https://www.Drug-PIN.com; request a free trial for research at hello@drug-pin.com). A Drug-PIN score of DDIs based on multiple patient drug interactions was performed for each patient. The medical Drug-PIN^®^ software allows to highlight drug interactions by combining them with demographic, clinical and biochemical data of patients. Thus, the Drug-PIN tool includes the full spectrum of variables influencing drug response: age, body mass index (BMI), race, kidney, and liver function, smoking habits, alcohol use, type and number of medications taken in chronic therapy and pharmacogenomics profile if available. In addition, the grade of interaction was evaluated according to the score obtained corresponding to the Drug-PIN traffic light and classified as low including green or yellow light (score 0–20 and 20–30, respectively) or high including dark yellow, orange, or red light (score 30–70, 60–70 or >70, respectively).

### 2.3. Statistical Analysis

Statistical analyses were performed using R statistical software (R: A Language and Environment for Statistical Computing. R Core Team, R Foundation for Statistical Computing, Vienna, Austria, https://www.R-project.org; V 4.3.1). Drug-PIN score and Drug-PIN traffic light variables were analyzed about toxicities for a cohort of 177 patients. To investigate the relationships between the toxicity variables and the drug interaction variables, a heatmap was built by calculating the Pearson correlation coefficients (and their corresponding *p*-values) among each pair of variables of toxicity with the drug interaction ones for all the analyzed patients. Then, the differences in Drug-PIN score and Drug-PIN traffic light variables were tested for statistical significance between groups of patients with and without different types of toxicities using the Student’s *t*-test. A *p*-value ≤ 0.05 was considered statistically significant. Chi-square (for large-sized samples) and Fisher’s exact tests (for small-sized samples) were also exploited to test the relationship between two classification factors, i.e., to assess for independence between Drug-PIN score and toxicity variables [49]. A *p*-value of 0.05 or less was considered statistically significant, meaning there is a relation between the two classification factors. To analyze the correlation between the Drug-PIN traffic light and toxicity variable with the patient OS and the PFS, the cumulative survival rates were computed according to the Kaplan–Meier (KM) method [50]. The survival outcomes of the different patient groups separated by Drug-PIN traffic light class (i.e., green or yellow versus dark yellow, orange or red) or by the presence of toxicity were compared by the median of the log-rank test, with a *p*-value ≤ 0.05 considered as statistically significant. A multiple linear regression analysis was then performed to assess the prediction of the cardiotoxicity outcome variable based on the combination of different predictor variables (e.g., age, Drug-PIN score or Drug-PIN traffic light, number of drugs, comorbidity, and performance status).

## 3. Results

### 3.1. Patients

In total, 177 patients with metastatic/unresectable BRAFV600 CM were enrolled in the study. Their clinical-pathological features are reported in Table 1.

The median age was 62 years (range 23–88 years), with 117 male patients (66.1%) and 60 female patients (33.9%). Baseline ECOG-PS, evaluated before starting the treatment, was 0, 1, and 2 in 103 (58.2%), 55 (31.1%), and 19 (10.7%) patients, respectively. At the baseline, 118 (66.6%) patients had comorbidities, among which the most frequent were as follows: cardiovascular comorbidities (including hypertension) in 71 patients, 19 patients with diabetes mellitus, 17 patients with a previous history of the oncological disease (including epitheliomas, basaliomas, urothelial carcinomas of the bladder, and prostate carcinoma), 9 patients with dyslipidemia, 7 patients with BPCO, 11 patients with benign prostatic hypertrophy, 11 patients with thyroid pathology, 7 patients with autoimmune disease, 5 patients with neurological disease, 3 patients with hepatopathy, and 6 patients with psychiatric disorders.

In total, 90 patients (50.8%) had lung metastasis, 83 patients (46.9%) had liver metastasis, and 99 patients (55.9%) had nodal spread. Furthermore, 44 patients had brain metastasis (24.8%) (Table 1). All patients were treated with BRAFi/MEKi therapy. Overall, 148 patients (83.6%) received the treatment in a first-line setting and 29 (16.4%) in a second-line setting. Moreover, 144 patients (81.3%) underwent D/T, 29 V/C (16.4%), and only 4 (2.3%) were treated with the most recently approved combination of E/B.

### 3.2. Toxicities

Toxicity of any grade during combination BRAF/MEK inhibitors treatment was registered in 79% of patients, with 140/177 patients experiencing at least one toxicity. In total, 29 patients needed dose reduction (16.4%). Treatment discontinuation for AE was reported in 15 patients (8.5%). The toxicities experienced in the study population are shown in Table 2.

The most frequent AEs were fever (66, 37.3%), skin reactions (66, 37.3%), asthenia (51, 51.8%), cardiovascular toxicity (31, 17.5%), diarrhea (25, 14%), and nausea (24, 13.6%). QT-interval (QT) prolongation at ECG was the most frequent cardiovascular side effect observed (16 patients). Heart rhythm disorders were registered in 10 patients: two cases of tachycardia, bradycardia, extrasystole, flutter, and atrial fibrillation. Intraventricular conduction disorders occurred in five patients. Myocardial infarction (one patient), acute heart failure (one patient), and pericardial effusion (two patients) occurred as well. Hypertension was detected in one patient treated with combo BRAFi/MEKi. More than one cardiac toxicity occurred concurrently in five patients. In total, 34 (19.2%) grade G3–G4 toxicities were diagnosed in the study population. Liver impairment (six cases), cardiotoxicity (five cases), renal impairment (four cases), and neutropenia (four cases) were the most significant severe toxicities reported (Table 2). Further, 5 out of 31 patients with cardiovascular toxicities had high-grade G3–G4 events. G3 toxicity included four cases with G3 QT-interval prolongation and the case in which acute heart failure occurred.

The cardiological toxicities are described in Table 3. median Drug-PIN score post-anti-BRAFi/MEKi therapy in patients with cardiovascular toxicity was 21.72.

### 3.3. Concomitant Medications and Drug-PIN Score

In total, 118 patients had comorbidities (66.7%) and 94 patients (55.9%) required specific pharmacological treatment, taking at least one concomitant medication, with an average of three drugs daily. Moreover, 22 patients took five or more daily drugs. The main classes of co-administered medications are listed in Table 4.

The median Drug-PIN score was 4.2 with a green (98), yellow (10), dark yellow (8), orange (1), and red (1) Drug-PIN traffic light before starting combined treatment with BRAF/MEK inhibitor. Additionally, 10 patients had a pre-BRAFi/MEKi therapy Drug-PIN traffic light ranging from dark yellow to red. In total, 8 of these 10 patients developed toxicity: 4 had cardiological toxicities (2 had G3 cardiological toxicities, 1 had G2, and 1 had G1).

The median Drug-PIN score increased to 5.44 when the target combination was added to the patient’s home therapy. A relevant change in Drug-PIN traffic light in 54 patients, with 13 patients moving to red light, 22 to yellow, and 19 to dark yellow, was also reported (Table 5).

Moreover, the addition of targeted therapy significantly contributed to increasing the risk of DDIs, according to the Drug-PIN score value moving from pre-BRAFi/MEKi therapy to BRAFi/MEKi therapy addiction (*p*-value < 0.0001) (Figure 1).

Here, 37 patients had a high grade of interaction after taking oncological treatment (Drug-PIN traffic light from dark yellow to red). In total, 30 patients developed toxicities during therapy, which were cardiological in 10 patients. Eight patients developed G3–G4 toxicities. In two of the eight patients who developed severe toxicities, the combination of antiarrhythmics and anticoagulants recurred (Table 6).

A tendency for increasing serious toxicities rate with higher Drug-PIN scores was noted, as seen in Figure 2A, although this finding was not statistically significant. Severe toxicities increased for higher median values of the Drug-PIN score without reaching the statistical significance threshold (*p* = 0.98, Figure 2B).

At the same time, the correlation between Drug-PIN score and AE toxicity grade was not statistically significant (*p*-value > 0.05 Figure 3A). When data were analyzed based on the median Drug-PIN score value (5.44), this trend was confirmed, although the data did not reach statistical significance (*p*-value = 0.97) (Figure 3B).

BRAFi/MEKi dose reduction was not to be associated with the Drug-PIN score. As shown in Figure 4A, dose reduction does not appear to correlate with drug interaction severity. Moreover, according to the Drug-PIN score’s median value, the dose reduction rate did not vary significantly (Figure 4B).

### 3.4. Cardiologic Toxicity

As can be seen from the correlation plot (Figure 5A), the only statistically significant finding is the association between the Drug-PIN score and traffic light and cardiological toxicity. Thus, it was found that the risk of cardiological adverse reactions increases as the Drug-PIN score increases (i.e., it is significantly associated with drug interactions). Cardiovascular toxicity was significantly related to the Drug-PIN score (*p*-value 0.048). Indeed, the risk of developing cardiological side effects appeared to correlate with DDIs and elevated Drug-PIN scores (Figure 5B,C).

Of 31 patients who developed cardiotoxicity, 23 had comorbidities, 16 had cardiological comorbidities at baseline, and 19 took at least one drug. In total, 12 patients suffered from hypertension, 3 had a history of ischemic heart disease, 4 had rhythm disturbances, 1 had chronic heart failure, and 1 reported ECG changes. Five patients had at least two cardiological comorbidities in their history. In total, 14 of the 16 patients took at least one home medication: 2 patients took exclusively non-cardiological drugs, and 13 patients took both cardiological (i.e., antihypertensive drugs, diuretics, antiplatelet, and anticoagulants) and non-cardiological medications. In this group of patients, Drug-PIN traffic light pre-anti-BRAFi/MEKi therapy was dark yellow in two, orange in one, and red in one patient. Post BRAFi/MEK inhibitors therapy, the Drug-PIN traffic light became red in five patients, suggesting an important impact of target therapy in drug interactions and cardiotoxicity. Out of 15 patients, 10 were taking sartans/ACE inhibitors, 5 were taking anti-platelets drugs, 3 were taking anticoagulants, 7 of them beta-blockers, 6 of them diuretics, 6 calcium antagonist, 3 were oral antidiabetics drugs, 2 antiarrhythmics (amiodarone and flecainide), 5 statins, 1 both an antidepressant belonging to the class of selective serotonin reuptake inhibitors (or SSRIs) and an antipsychotic neuroleptic drug, and 2 were taking benzodiazepines. In total, 15 of 31 patients had no cardiological comorbidity, but 4 took concomitant medications at the baseline for other reasons. One patient took sodium valproate and benzodiazepines, while the other was under treatment with levetiracetam, alfuzosin, and levothyroxine. The remaining two patients were taking heparin and statin only, respectively. In this group, Drug-PIN light was green at the baseline in all patients, but post-BRAF/MEK inhibitors therapy, it became yellow in three patients and dark yellow in two patients. Therefore, in this group of patients, cardiological toxicity could be attributed mainly to the toxicity profile of BRAFi/MEKi therapy (Figure 6).

### 3.5. Multiple Regression Analysis: Drug-PIN and Cardiologic Toxicities

Considering several clinical parameters, including age, gender, performance status, comorbidity, number of drugs taken in concomitant therapy, and Drug-PIN, we observed that only the Drug-PIN traffic light variable or the Drug-PIN score are considered as predictors for cardiotoxicity (Table 7 and Table 8, respectively).

### 3.6. Survival Analysis

Overall, developing drug toxicity did not result from having an impact on survival outcomes. No significant difference was detected in OS and PFS between patients with non-severe vs. severe toxicities, irrespective of whether the toxicities were related or not to drug interactions (Figure 7).

To assess the impact of Drug-PIN on OS and PFS outcomes, we performed a survival analysis based on green–yellow vs. dark yellow–orange-red Drug-PIN traffic light. Patients in the green or yellow classes showed a good prognosis compared to those in the dark yellow, orange, or red classes, both in terms of OS and PFS (Figure 8).

Considering only the subgroup of patients with cardiological toxicity, the survival analysis showed that patients with a green or yellow Drug-PIN traffic light had better outcomes both in terms of OS and PFS compared to the ones with the dark yellow, orange, or red Drug-PIN light, as shown in Figure 9.

## 4. Discussion

Our study highlights that, in patients with advanced melanoma treated with BRAFi/MEKi, the presence of DDIs favors the development of adverse drug reactions; in particular, DDIs appear to be significantly associated with an increased risk of cardiological toxicity. Moreover, DDIs also have a prognostic value as they are associated with worse oncological outcomes in terms of OS and PFS, probably because they affect both the efficacy of anti-BRAFi/MEKi drugs and treatment adherence.

As expected, most patients (66.7%) had comorbidities, 99 of which (55.9%) required a specific concomitant pharmacological treatment. Thus, more than half of the patients were exposed to complex drug regimens, which increased the risk of potential drug–drug interactions and, consequently, reduced the efficacy and safety of oncological therapies. Patients were at risk of being undertreated for their comorbidities since concomitant treatment may be reduced by DDIs [51]. It is worthy of note that ten patients had potentially dangerous DDIs (dark yellow (8), orange (1), and red (1) Drug-PIN light) before the start of BRAFi/MEKi treatment. Furthermore, most of these patients (8/10) developed toxicity when BRAFi/MEKi treatment was added.

Interestingly, cardiological toxicity occurred in half of these cases, confirming the central role of DDIs in the development of AEs and the onset of cardiotoxicity. In this scenario, adding BRAFi/MEKi treatment contributed to the increase in the DDIs score. When the treatment with BRAFi/MEKi was added, the median Drug-PIN score increased from the basal score of 4.2 to 5.4 (*p*-value < 0.0001). A relevant change in the Drug-PIN traffic light was also reported in 54 patients, with an increase in potentially dangerous interactions. The increased risk of toxicities related to DDIs could translate to a critical clinical impact, especially when the patients are exposed to a treatment burdened by a high toxicity profile. In phase III clinical trials, related adverse effects were experienced in nearly all patients (97%) treated with BRAFi/MEKi [29]. During BRAFi/MEKi treatment, 78% of the patients experienced toxicity of any grade. This lower-than-expected incidence of AEs may be partially related to lower reports of low-grade toxicities, requiring no action or dose adjustments in clinical practice compared to clinical trials. However, pathologies currently directly linked to BRAFi/MEKi treatment, such as arterial hypertension, were treated pharmacologically and monitored before therapy in all patients. Antihypertensive treatment was started or pursued according to the existing guidelines. This proactive attitude could partially explain the very low rate of newly-onset arterial hypertension reported compared to that evidenced in clinical trials (ranging from 11% to 29%). High-grade toxicities, occurring in 19.2% (34 patients), also had a lower rate [29]. Nevertheless, their management required dose reduction and treatment discontinuation in 16.6% and 8.5% of cases, respectively. It should also be stressed that the toxicity profile was similar to that emerging from clinical trials, with no unexpected toxicity, even in patients with a high DDI score.

This study is not exempt from limitations. It failed to demonstrate that higher Drug-PIN scores and traffic lights are related to a higher number of toxicities of all types, high-grade toxicities, unexpected toxicities, and dose reduction. Nevertheless, even when the data did not reach statistical significance, patients with higher Drug-PIN scores tended to experience more toxicities in this series. Any significant associations between toxicity and the number of concomitant medications were found, suggesting that the type of interactions between drugs is more relevant than the number of interactions in the onset of the toxicity profile. Conversely, this study evidenced that a high Drug-PIN score was significantly associated with cardiovascular toxicity, predominantly constituted by acute or subacute clinical manifestations; more frequently, they were represented by disturbances of the repolarization, QT-interval prolongation, ventricular and supraventricular arrhythmias, conduction disorders and acute heart failure. Multiple regression analysis also confirmed the predictive role of Drug-PIN traffic light and score concerning the onset of cardiological toxicity, compared to several other clinical parameters (age, gender, performance status, comorbidity, and several concomitant drugs).

Cytochromes of the P450 superfamily of enzymes (CYPs) play a central role in the metabolism of BRAF-i/MEK-i drugs and drug–drug interaction onset. Numerous classes of drugs (cardiological and non), cytochromes inhibitors or inducers, can influence plasma concentrations of drugs and consequently treatment efficacy, related toxicity profile, and treatment adherence.

Both BRAFi and MEKi showed a cardiovascular toxicity profile and cardiovascular toxicity, which is more frequent when they are used in a combination therapy rather than in a monotherapy setting [52]. BRAFi treatment is associated with hypertension and QT-interval prolongation, while hypertension, peripheral edema, and cardiomyopathy with decreased cardiac ejection fraction have been usually associated with MEKi treatment [53,54,55,56]. Cardiovascular AEs may be directly related to the effect that BRAFi and MEKi exert on the selective inhibition of the pathway in the heart. Cardiovascular side effects can be considered an on-target effect of BRAFi/MEKi treatment. In 15/31 patients, those who did not have overt cardiological comorbidities, did not take drugs with a cardiotoxic profile, and did not present any high DDI, cardiotoxicity onset could be considered predominantly related to the cardiotoxic effect of the BRAFi/MEKi treatments.

A second mechanism for the onset of cardiotoxicity could be ascribed to a synergistic effect between drugs, which have the same cardiac adverse effect, QT prolongation occurring in the 16 patients (9%) in our study. QT prolongation was observed in up to 3–7% of patients treated with V and 2% treated with V+C. Grade ≥ 3 QT-interval prolongation occurred in 1% of patients treated with V monotherapy or V+C. Due to an additional fluorinated phenyl ring, negative effects on QT prolongation were not seen with D or E. In this series, high-grade QT prolongation was observed in four patients (2.3%). This drug interaction could lead to early depolarizations, triggering the initiation of torsade de pointes ventricular tachycardia. A potential synergist effect related to DDI was underlined in QT prolongation during BRAFi/MEKi treatment in 15 patients who did not have cardiovascular disease at baseline and developed cardiotoxicity. The involved medication could be any other QT-prolonging drug (e.g., pantoprazole, antibacterial and antifungal agent ciprofloxacin) and ethanol, antiemetic and prokinetic agents, ondansetron, and psychotropic agents such as tricyclic antidepressants, citalopram, escitalopram, haloperidol, methadone, pimozide, and thioridazine, which were taken concomitantly enhancing DDI scores [51].

In addition, the MAPK pathway in cardiomyocytes is a protective signaling pathway, and its pharmacological inhibition interferes with intramyocytic repair mechanisms by inhibiting the extracellular signal-regulated kinases ERK 1⁄2 [51,56,57,58,59]. An impaired protective effect of the pathway may expose the myocardium more easily to damage. In this scenario, concomitant medications with high/intermediate DDIs may mediate subclinical cardiotoxicity or damage until significant left ventricular systolic dysfunction or even heart failure induced by the co-exposure of drugs with a potentially cardiological toxicity profile ensue. Non-steroidal anti-inflammatory drugs (NSAIDs) consumption for pain management may increase the risk of myocardial infarction [60]. The increased risk is attributed to the imbalance caused by the inhibition of COX-2 cyclooxygenases and prostacyclin without inhibition of COX-1 and, therefore, thromboxane. They are also connected with platelet inhibition, oxidative stress, and impaired regulation of renal perfusion.

Similarly, some neurological and psychiatric drugs, cortisones, bisphosphonates, and some antibiotics (erythromycin) may have a potential cardiological toxicity profile that could become critical when intramyocyte repair mechanisms are impaired. Furthermore, a DDIs-dependent reduced availability of cardiac drugs may lead, in patients requiring concomitant cardiac treatment, to an under treatment of the cardiac comorbidity, which could become critical when cardiomyocytes exhibit pharmacological inhibition of the cardioprotective pathway of MEK. DDIs in these patients increased the risk for significant left ventricular systolic dysfunction or even heart failure induced by the concomitant or subsequent use of BRAFi and MEKi.

In cancer patients with concomitant cardiovascular pathologies, left ventricular dysfunction, ranging from asymptomatic changes (best diagnosed by echocardiographic strain analysis) to severe cardiac failure, is related to several factors. Both metabolic (diabetes, arterial hypertension, arterial disease, metabolic syndrome/obesity, BMI > 30 kg/m^2^, etc.) and cardiological (genetic factors, age > 65 years, gender, underlying impaired myocardial function, and vascular status) comorbidities increase the risk of impaired myocardial function during treatment. Even here, the difference between therapeutic doses and doses evoking DDIs-related deleterious effects might not be clear, and the interindividual susceptibility difference is high. Furthermore, previous treatments, including immunotherapy-mediated subclinical cardiotoxicity or damage induced by radiotherapy, may enhance the risk of developing cardiotoxicity during therapy with BRAF/MEK inhibitors. Notably, other factors occurring during treatment, like electrolyte imbalance due to diarrhea, stomatitis, or long QT syndrome, could potentiate the AEs and should be promptly corrected [29]. Therefore, the risks and benefits of BRAFi/MEKi therapy should be carefully evaluated in patients with significant heart disease and managed in a multidisciplinary consensus.

This study evidenced that DDI must be considered an additional risk factor for cardiological adverse reaction onset in this complex and multifactorial scenario. The risk of cardiological adverse reactions during treatment with BRAFi/MEKi increases when the Drug-PIN score rises, and the risk of cardiovascular AEs is significantly associated with drug interactions. Therefore, any DDIs must be assessed before cancer therapy, and then drug–drug interactions promptly removed when possible. It is crucial to remove an additional cardiac toxicity risk factor and promptly detect left ventricular dysfunction during treatment with BRAF/MEK inhibitors since it often results in discontinuation of the oncological therapy and, while most of the cardiac side effects could be adequately managed and are reversible with the interruption of treatment, fatal events considered to be due to arrhythmias or sudden cardiac death may occur.

The presence of high-grade DDIs in approximately 30.5% (54 patients with Drug-PIN traffic light from dark yellow to red) of patients receiving BRAFi/MEKi does correlate with a significant reduction in OS and PFS. Survival analysis showed that in the overall population, a green-yellow Drug-PIN traffic light correlates with a better OS and PFS than a dark yellow, orange, or red Drug-PIN one. In addition, considering only patients who develop cardiological toxicity, the green-yellow Drug-PIN traffic light correlates with a better OS and PFS, confirming that a low number DDIs is likely to allow better adherence to treatment and ensure better efficacy of anti-BRAFi/MEKi therapy. On the other hand, the presence of any grade toxicity does not correlate with both OS and PFS, in contrast with the available literature. Recent evidence suggests that the development of immune-related toxicities such as vitiligo, keratitis, uveitis, and erythema nodosum under BRAFi/MEKi could be associated with long-term benefits in terms of oncological outcomes [61].

The study’s main limitation is its retrospective nature, which only allows for a more in-depth investigation of some aspects of drug interactions. Indeed, these results suggest that the Drug-PIN may have a significant prognostic and predictive value for oncological outcomes and the development of severe toxicities that should be further investigated in prospective, targeted studies.

## 5. Conclusions

In conclusion, this series of 177 BRAF-mutant MM patients treated with combined BRAFi/MEKi provides several important indications regarding the impact of DDIs on the toxicity profile of the treatment, offering insight into drug interactions in a real-world population. The assessment of potential DDI, the use of alternative medications when possible, and the careful monitoring of the toxicities must become mandatory since the risk of drug interactions in clinical practice is consistent. The impact on oncological outcomes and the toxicity spectrum of patients with BRAFv600mutant MM treated with BRAFi/MEKi emerges as relevant and deserving of further investigation. Knowledge and removal of DDIs are critical to ensure the best adherence to oncological treatment and minimize the related toxicities, including the significantly increased risk of cardiovascular toxicity, which appeared as a crucial safety issue and a theme of major concern in precision medical oncology.

## Figures and Tables

**Figure 1 cancers-15-04587-f001:**
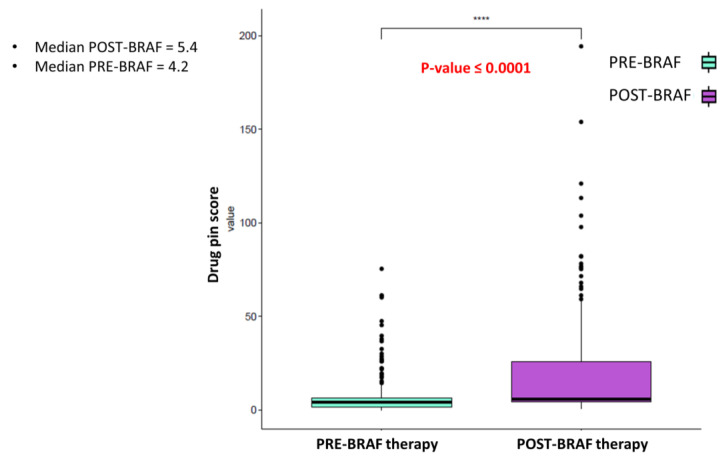
Boxplot of the Drug-PIN score variable in 177 patients pre and post-BRAF therapy. The *p*-value was obtained by performing a Student *t*-test. The addition of anti-BRAFi/MEKi therapy significantly increases the risk of DDIs. **** indicates *p*-value ≤0.0001.

**Figure 2 cancers-15-04587-f002:**
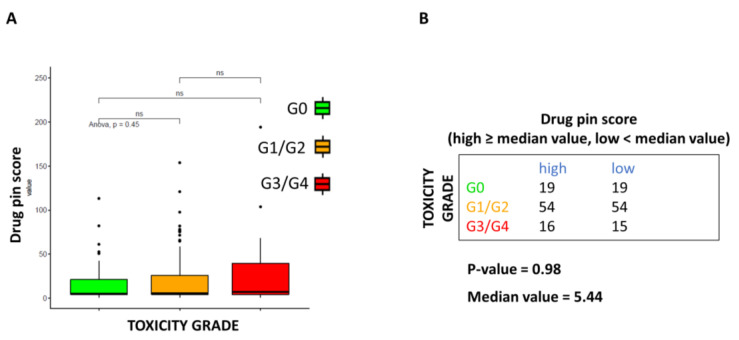
Correlation between Drug-PIN score and toxicities grade. (**A**) Boxplot of the Drug-PIN score variable in 38 patients without toxicity (green box), 108 with toxicity G1-2 (equal to 1, orange box), and 31 patients with toxicity G3–4 (equal to 2, red box). Pairwise *p*-values were obtained by performing a Student *t*-test, and multiple comparisons were performed using the ANOVA test. (**B**) Frequency table for the two classifications: toxicity grade (rows) and Drug-PIN score (columns). In particular, the median value of the Drug-PIN score (i.e., corresponding to 5.44) was defined as a cut-off to split the patients into two groups: one group showing a Drug-PIN score value higher than or equal to the selected cut-off (high), and another group showing a Drug-PIN score value lower than the chosen cut-off (low). χ2 test was performed to test the relationship between the two classification factors. ns: indicates a non-significant statistical difference.

**Figure 3 cancers-15-04587-f003:**
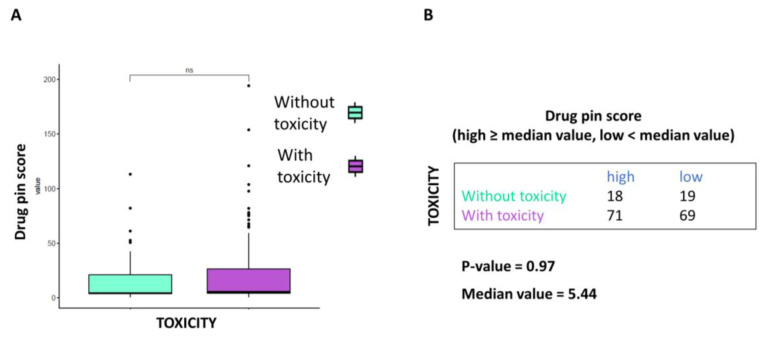
Correlation between Drug-PIN score and toxicities of any grade. (**A**) Boxplot of the Drug-PIN score variable in 37 patients without toxicity (water blue box) and 140 patients with any grade toxicity (violet box). The *p*-value was found by performing the Student *t*-test. (**B**) Frequency table for the two classification factors: toxicity (rows) and Drug-PIN score (column). In particular, the median value of the Drug-PIN score (i.e., corresponding to 5.44) was defined as a cut-off to split the patients into two groups: one group showing a Drug-PIN score value higher than or equal to the selected cut-off (high), and another group showing a Drug-PIN score value lower than the chosen cut-off (low). *p*-value was obtained by performing a chi-square to test the relationship between the two classification factors. ns: indicates a non-significant statistical difference.

**Figure 4 cancers-15-04587-f004:**
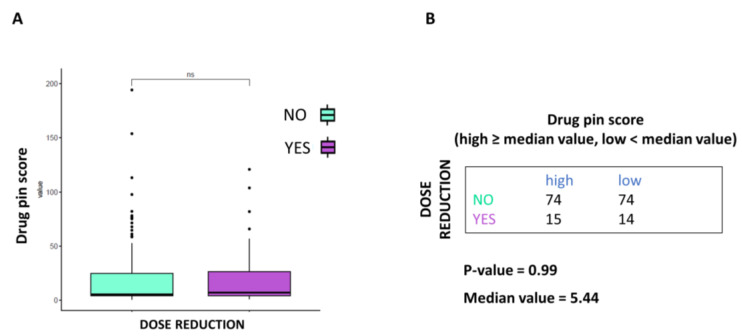
Correlation between Drug-PIN score and dose reduction. (**A**) Boxplot of the Drug-PIN score variable in 148 patients without dosage reduction (water blue box) and 29 patients with dosage reduction (violet box). The *p*-value was obtained by performing a Student *t*-test. (**B**) Frequency table for the two classification factors: dosage reduction (rows) and Drug-PIN score (column). In particular, the median value of the Drug-PIN score (i.e., corresponding to 5.44) was defined as the cut-off to split the patients into two groups: one group showing a Drug-PIN score value higher than or equal to the selected cut-off (high), and another group showing a Drug-PIN score value lower than the chosen cut-off (low). *p*-value was obtained by performing a chi-square to test the relationship between the two classification factors. ns: indicates a non-significant statistical difference.

**Figure 5 cancers-15-04587-f005:**
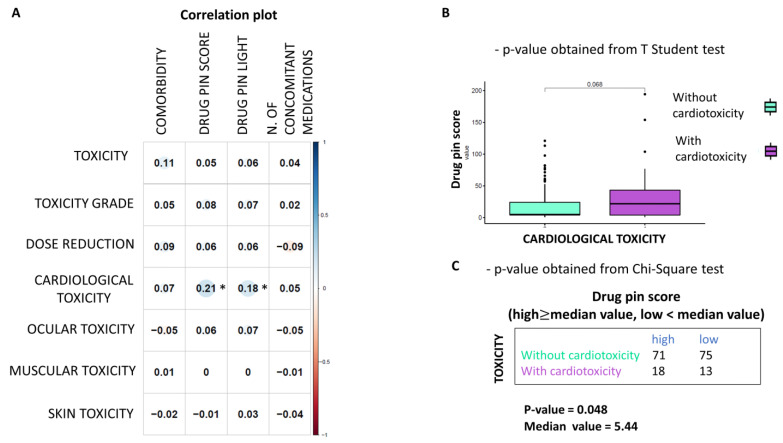
Correlation between Drug-PIN score and cardiological toxicities (**A**) Heatmap of Pearson correlation coefficients between the toxicity values (row) and the other variables (columns) are reported. The plot’s circles are scaled and colored according to the correlation values, increasing from red (negative correlation) to blue (positive correlation). A star marks statistically significant *p*-values (*p* ≤ 0.05). Drug-PIN Score and Light have a positive and significant correlation with cardiological toxicity (**B**) Boxplot of the Drug-PIN scores variable in 31 patients with cardiological toxicity (violet box) and 146 patients without cardiological toxicity (water blue box). The *p*-value was obtained by performing a Student *t*-test. (**C**) Frequency table for the two classification factors: cardiological toxicity (rows) and Drug-PIN score (column). In particular, the median value of the Drug-PIN score (i.e., corresponding to 5.44) was defined as the cut-off to split the patients into two groups: one group showing a Drug-PIN score value higher than the selected cut-off (high), and another group showing a Drug-PIN score value lower than the chosen cut-off (low). *p*-value was obtained by performing a chi-square to test the relationship between the two classification factors.

**Figure 6 cancers-15-04587-f006:**
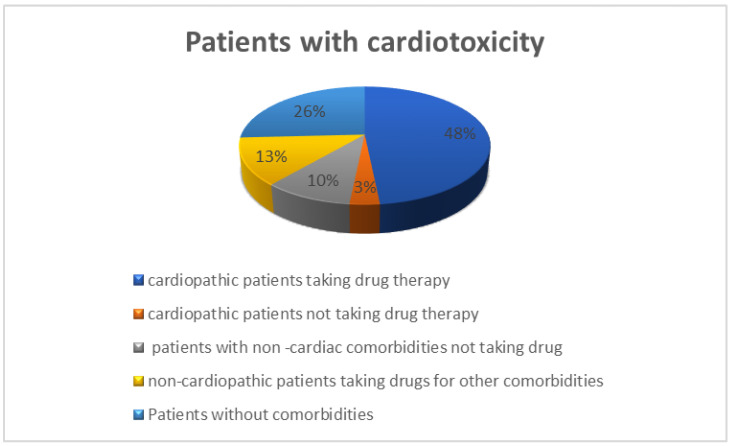
Distribution of patients with cardiological toxicity in relation to comorbidities and their drug history 15 cardiopathic patients taking specific drug therapy (48%), one cardiopathic patient not taking medications (3%), eight patients without comorbidities not taking drugs (26%), four non-cardiopathic patients taking medications for other comorbidities (13%), three patients with non-cardiac comorbidities not taking drugs (10%).

**Figure 7 cancers-15-04587-f007:**
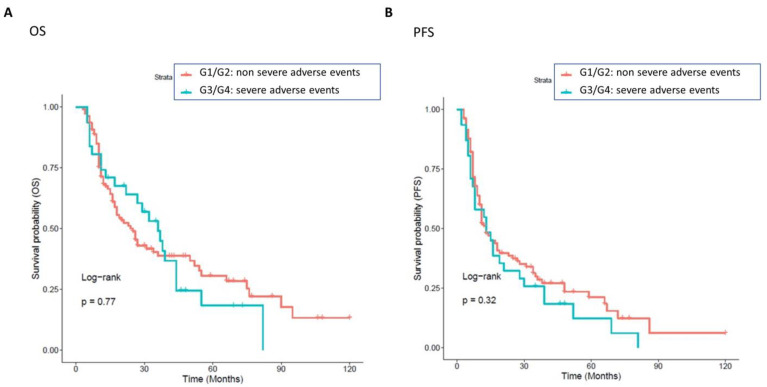
Association between non-severe and severe toxicities, with OS and PFS. Patients with a grade of toxicities greater than 0 were classified into two groups: one class including patients with severe toxicity (cyan curve, grade G2, 31 patients), and the other class including patients with a non-severe grade of toxicity (red curve, grade G1, 108 patients). The correlation between variable value and patient survival was examined as overall survival (OS) (panel (**A**)) and progression-free survival (PFS) (panel (**B**)). The prognosis of each group of patients was examined by Kaplan–Meier survival estimators, and the survival outcomes of the two groups were compared by log-rank tests. Log-rank *p*-values less than or equal to 0.05 were considered statistically significant. No significant differences were found in OS and PFS in patients with non-severe (red curve) or without severe (cyan curve) grades of toxicities.

**Figure 8 cancers-15-04587-f008:**
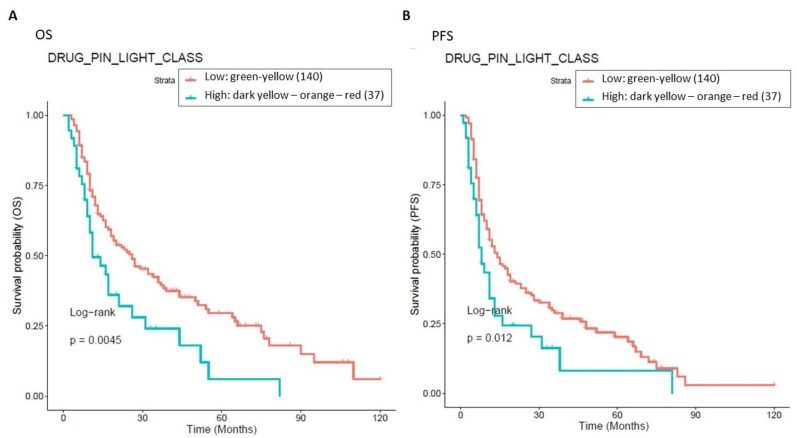
Association between OS, PFS, and Drug-PIN traffic light. 177 patients were classified into two groups: one class including patients with a Drug-PIN traffic light equal to green or yellow (136 patients, cyan curve), and the other one including patients with a Drug-PIN traffic light equal to dark yellow, orange, or red (37 patients, red curve). The correlation between variable value and patient survival was examined as OS (panel (**A**)) and PFS (panel (**B**)). The prognosis of each group of patients was analyzed by Kaplan–Meier survival estimators, and the survival outcomes of the two groups were compared by log-rank tests. Log-rank *p*-values ≤ 0.05 were considered to be statistically significant.

**Figure 9 cancers-15-04587-f009:**
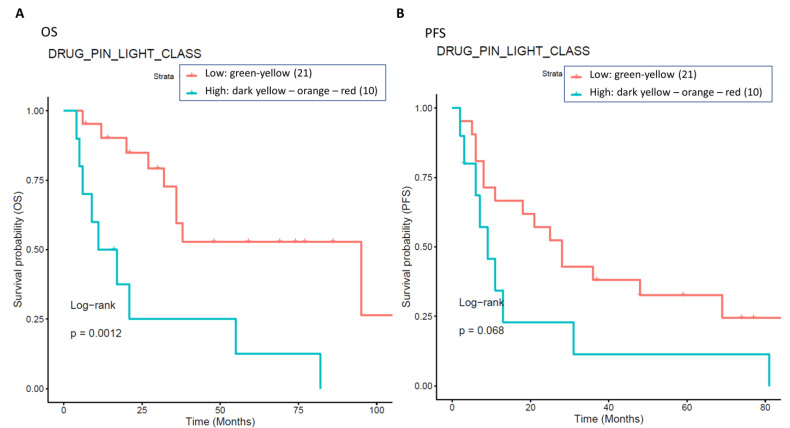
The impact of Drug-PIN on survival in a subgroup of patients with cardiological toxicities. Thirty-one patients with cardiological toxicity were classified into two groups: one class including patients with Drug-PIN traffic light variable equal to dark yellow, orange, or red (10 patients, cyan curve), and the other one including patients with Drug-PIN traffic light variable equal to green or yellow (20 patients, red curve). The correlation between variable value and patient survival was examined as OS (panel (**A**)) and PFS (panel (**B**)). The prognosis of each group of patients was analyzed by Kaplan–Meier survival estimators, and the survival outcomes of the two groups were compared by log-rank tests. Log-rank *p*-values ≤ 0.05 were considered statistically significant.

**Table 1 cancers-15-04587-t001:** Clinical and pathological characteristics.

	All Patients *n* 177 (%)
**Age (years)**	
Median age (range)	62 (23–88)
**Gender**	
Male	117 (66.1)
Female	60 (33.9)
**Performance status**	
0	103 (58.2)
1	55 (31.1)
2	19 (10.7)
**Comorbidity**	
Yes	118 (66.6)
No	59 (33.4)
**Line of treatment anti-BRAF**	
I	148 (83.6)
II	28 (15.8)
III	1 (0.6)
**Site of metastasis at baseline**	
Lymph nodes	99 (55.9)
Lung	90 (50.8)
Liver	83 (46.9)
Soft tissue	50 (28.2)
Brain	44 (24.8)
Bone	34 (19.2)
Skin	32 (18.0)
Peritoneum	14 (7.9)
Locoregional recurrence	6 (3.4)
Pleura	4 (2.2)
**Type of anti-BRAF Therapy**	
Dabrafenib/Trametinib	144 (81.3)
Vemurafenib/Cobimetinib	29 (16.4)
Encorafenib/Binimetinib	4 (2.3)

**Table 2 cancers-15-04587-t002:** Type of toxicities occurred and their grading.

Toxicities	Patients (%)	G1–G2	G3–G4
Pyrexia	66 (37.3)	66	-
Skin Toxicity	66 (37.3)	63	3
Asthenia	51 (28.8)	48	3
Cardiological	31 (17.5)	26	5
Diarrhea	25 (14.1)	24	1
Nausea	24 (13.6)	24	-
Liver Function	24 (13.6)	18	6
Muscle Toxicity	22 (12.4)	21	1
Renal Function	10 (5.6)	6	4
Neutropenia	10 (5.6)	6	4
Ocular Toxicity	9 (5.0)	8	1
Anaemia	9 (5.0)	6	3
Mucositis	6 (3.4)	4	2
Constipation	6 (3.4)	5	1
Arthralgias	6 (3.4)	6	-

**Table 3 cancers-15-04587-t003:** Type of cardiological toxicity and grading.

Cardiotoxicity	Patients	G1–G2	G3–G4
QT-Interval Prolongation	16	12	4
Tachycardia	2	2	-
Bradycardia	2	2	-
Extrasystole	2	2	-
Right Branch Block	2	2	-
Intraventricular Conduction Disorders	3	3	-
Flutter	2	1	-
Atrial Fibrillation	2	2	-
Acute Myocardial Infarction	1	1	-
Pericardial Effusion	2	2	-
Hypertension	1	1	
Acute Heart Failure	1	-	1

**Table 4 cancers-15-04587-t004:** Classes of concomitant drugs registered at the baseline.

Class of Drug	Patients
Ace Inhibitors	37
Diuretics	20
Beta-Blockers	20
Sartans	19
Calcium Antagonist	18
Antiplatelet	17
Proton Pump Inhibitors	13
Statins	12
Oral Antidiabetics	11
Corticosteroids	10
Benzodiazepine	9
Thyroid Hormone	9
Insulin	9
Alfa-Blockers	8
Antiepileptic	7
Antidepressants	4
Anticoagulants	3
Antiarrhythmic	3
Anti-Viral	2
Low Molecular Weight Heparin	1

**Table 5 cancers-15-04587-t005:** Changes in Drug-PIN traffic light before and after BRAFi/MEKi therapy.

Drug-Pin Light	Pre-Anti-BRAF/MEK	Post-Anti-BRAF/MEK
Green	98	62
Yellow	10	22
Dark Yellow	8	19
Orange	1	2
Red	1	13

**Table 6 cancers-15-04587-t006:** Severe toxicities in patients with high-grade interactions.

Toxicity	Patients (N)	G3	G4
Skin Toxicity	1	1	-
Muscle Toxicity	1	1	-
Cardiological Toxicity	3	2	1
Asthenia	2	2	
Impairment of Liver Function	1	1	-
Anaemia	1	1	-
Diarrhea	1	1	-

**Table 7 cancers-15-04587-t007:** Multiple regression analysis of cardiotoxicity: Drug-PIN score.

Parameter	Coefficient	Standard Error	T-Value	*p*-Value
N Concomitant Medications	−0.077277	0.084103	0.919	0.35952
**Drug-Pin Score**	**0.002932**	**0.001096**	**2.675**	**0.00821 ****
Age	0.002500	0.002191	1.141	0.25544
PS	0.002267	0.072841	0.031	0.97521
Comorbidity	0.039110	0.085565	0.457	0.64821
Gender	0.081196	0.060482	1.342	0.18127

PS: performance status; in bold variables that reached *p*-value < 0.05; ** *p*-value < 0.01.

**Table 8 cancers-15-04587-t008:** Multiple regression analysis of cardiotoxicity: Drug-PIN traffic light.

Parameter	Coefficient	Standard Error	T-Value	*p*-Value
N Concomitant Medications	−0.0709705	0.0858316	0.827	0.4095
Drug-PIN Traffic Ligh	**0.0621005**	**0.0282032**	**2.202**	**0.0291 ***
Age	0.0023763	0.0022236	1.069	0.286
PS	0.0004244	0.0735392	0.006	0.9954
Comorbidity	0.0430719	0.0862697	0.499	0.6182
Gender	0.0778916	0.0608578	1.280	0.2024

PS: performance status; in bold variables that reached *p*-value < 0.05; * *p*-value < 0.05.

## Data Availability

Individual participant data that underlie the results reported in this article, after de-identification, study protocol, informed consent form, and statistical analysis plan will be available immediately following publication, with no end date, with researchers who provide a methodologically sound proposal, to achieve aims in the approved proposal. Proposals should be directed to simone.scagnoli@uniroma1.it to gain access; data requestors must sign a data access agreement.

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
