# Peer review of "The Impact of Drug–Drug Interactions on the Toxicity Profile of Combined Treatment with BRAF and MEK Inhibitors in Patients with BRAF-Mutated Metastatic Melanoma"

_cancers, 2023, doi:10.3390/cancers15184587_

Round 1

Reviewer 1 Report

The authors explored toxicity resulting from drug interactions in the patients receiving targeted inhibitor therapy for metastatic melanoma. This is crucial for risk assessment in choosing effective therapy and for managing regimens to minimize the risk of adverse effects or outcomes.

Overall the paper is well written and generates insights crucial for predicting adverse physiological effects of combination inhibitor therapies in melanoma. It also inspires mechanistic study into the biology of complex inhibitor drug interactions and the effect of simultaneously targeting two or multiple components of the MAPK pathway. The design is robust and well-powered.

Issues:

The procedure or methodology for computing drug PIN scores is not clearly defined. It is my understanding that the software is proprietary but a brief statement on variables fed into the complex algorithm calculating this score would be helpful without needing to visit the website.

Inconsistencies in the writing of "p-value"

Site of Metastasis (Table 1) should be ordered from highest frequency to least.

It is interesting not to observe any significant correlations/associations between toxicity and the number of concomitant medications.  Could there be an explanation for this?

Can R2 values be reported within Fig 5A?  This is important for gauging the strength of association/correlation beyond p-values.

“Extrasystole” is repeated twice in Table 3

The image resolution in Figure 5 is low. Figures and their legends and axis labels are small and difficult for readers.

Reviewer 2 Report

Comments for the authors

This multicenter and retrospective study describes the role of DDIs on the toxicity profile and clinical outcomes in a population of patients with metastatic melanoma showing BRAFV600 mutation and treated with BRAFi / MEKi therapy. 

The objective is to define the risk of drug interactions and their impact on the adverse events affecting drug tolerability of treatments. 

The clinical parameter assessment of the patients was matched with the DDI using the Drug-Personalized Interactions Network (PIN), a medical software able to detect drug interactions with patient profiles with their clinical and biochemical data. 

 In this retrospective study, the authors enrolled 177 patients. The clinical and pathological characteristics, the type of anti-BRAF combination therapy, the class of concomitant medication, grade, and the somewhat of toxicity during combination therapy were registered and evaluated. 

A drug PIN score of DDIs for every patient was associated and classified as low (green or yellow) versus high (dark yellow, orange, or red) before and after the BRAf/MEK therapy. 

The results indicate that, according to the Drug-PIN score value, the association therapy significantly increased the risk of DDIs. Toxicity of any grade during combination BRAF / MEK inhibitor treatment was registered and indicated with a change in the Drug-PIN traffic light.

Notably, the authors demonstrate a significative association between treatment with BRAF/MEK inhibitor and cardiovascular toxicity, present in 31 patients. In this group of patients, the Drug-PIN traffic light before and after treatment changed in the worse sense, suggesting the impact of target therapy in drug interactions causing cardiotoxicity. This change is well defined in fifteen patients with no cardiological comorbidity with Drug-PIN light green before treatment that became yellow in 3 patients and dark yellow in 2 patients, thus attributing the increased toxicity profile only to BRAFi/MEKi treatment.

The authors found that the survival analysis indicated as OS of patients with cardiotoxicity, was significantly better when the Drug-PIN traffic light was green or yellow compared to patients with orange or red Drug-PIN traffic light. In the PFS, a trend in the same direction was evident but not significant (log-rank tests >0,05).

This reviewer considers this study relevant in the target therapy area, eventually favoring better management of metastatic melanoma patients. The possible drug interactions could impact the result of treatments. The optimization of toxicity monitoring, in particularcardiotoxicity is mandatory for maximal adherence to the oncological treatment protocols, avoiding a change of the therapy, reduction of drug concentration, and eventually the possible exclusion of the patient from the therapeutic protocol.

 In general, the study is well-defined and presented.

The statistical approach is robust and corroborates the results 

Minor concern

 Increase the characters of Figure 5.

Author Response

We wish to thank the reviewer for his/her positive evaluation and comments.

 Figure 5 has been improved in resolution as suggested.